# The Impact of Renal Function on Hepatic Encephalopathy Following TIPS Placement for Refractory Ascites

**DOI:** 10.3390/biomedicines11082171

**Published:** 2023-08-02

**Authors:** Matthew Zhao, Sammy Saab, Chloe Craw, Edward Wolfgang Lee

**Affiliations:** 1Division of Hepatology, Department of Medicine, UCLA Medical Center, David Geffen School of Medicine at UCLA, Los Angeles, CA 90095, USA; myzhao@mednet.ucla.edu (M.Z.); ssaab@mednet.ucla.edu (S.S.); chloecraw@cox.net (C.C.); 2Division of Liver and Pancreas Transplantation Surgery, Department of Surgery, UCLA Medical Center, David Geffen School of Medicine at UCLA, Los Angeles, CA 90095, USA; 3Division of Interventional Radiology, Department of Radiology, UCLA Medical Center, David Geffen School of Medicine at UCLA, Los Angeles, CA 90095, USA

**Keywords:** transjugular intrahepatic portosystemic shunt, TIPS, chronic kidney disease, CKD, hepatic encephalopathy, HE, hemodialysis

## Abstract

Background: The impact of renal function on hepatic encephalopathy (HE) following transjugular intrahepatic portosystemic shunt (TIPS) placement for refractory ascites is poorly understood. We investigated the role of renal function on HE following TIPS placement. Methods: A retrospective study was performed for patients undergoing TIPS for refractory ascites from 2007–2019. Patients were stratified by GFR at time of TIPS placement and by whether they were on hemodialysis (HD). Chronic kidney disease (CKD) stage 3 or higher was defined as pre-TIPS GFR < 60 for at least 3 months. Logistic regression analyses were used to identify the role of GFR and CKD at time of TIPS placement on HE within 60 days post TIPS placement. Results: Among 201 TIPS patients for refractory ascites (61% male; mean age 59.1), 78 (39%) patients were in CKD, and 16 (21%) were on HD. Mean GFR at time of TIPS placement was 62.7 ± 28.2 for all non-HD patients (*n* = 185). Compared with the GFR ≥ 90 group, GFR < 30 or HD (OR, 3.56; 95%CI, 1.19–10.7; *p* = 0.023) and CKD (OR, 2.52; 95%CI, 1.40–4.53; *p* = 0.002) at time of TIPS placement were significant predictors of post-TIPS placement HE within 60 days. GFRs between 30–60 and 60–90 were not significant predictors. Conclusions: In TIPS patients for recurrent ascites, patients with acutely impaired renal function or chronic renal dysfunction were at an increased risk for HE after TIPS.

## 1. Introduction

Portal hypertension is a consequence of end-stage liver disease that manifests as variceal bleeding, ascites, and hepatic encephalopathy (HE). The development of ascites in these patients with chronic liver disease continues to represent one of the most common complications of portal hypertension [1]. Patients with diuretic-resistant ascites, as well as those that cannot tolerate it or have failed paracentesis, are generally considered for a transjugular intrahepatic portosystemic shunt (TIPS) and ultimately liver transplantation. A TIPS is therefore a crucial therapeutic modality for the treatment of ascites in selective patients with both compensated and decompensated liver disease. Nevertheless, known complications of TIPSs, including the precipitation of HE and worsening liver function, can be extremely distressing for patients and their family. 

Although TIPS placement is recommended for reducing the morbidity and progression of refractory ascites, examining outcomes among patients with end-stage liver disease and comorbid renal dysfunction remains an important area for continued study [2,3,4]. Superimposed renal dysfunction complicates the clinical management of patients with end-stage liver disease [5]. Moreover, patients with chronic liver disease are notably at increased risk of renal insufficiency due to worsening portal hypertension and the use of diuretics [6,7]. Renal function in the setting of existing liver disease also appears to be related to the development and resolution of HE [8,9,10].

Existing studies examining post-TIPS outcomes suggest that TIPS placement in patients with chronic kidney disease (CKD) is safe and efficacious, albeit with notably increased rates of postprocedural HE among TIPS recipients with CKD [11,12]. These previous studies, however, are limited in scope and do not control for the indication of TIPS placement. The proposed study will examine the role of both acute and chronic renal dysfunction on the development of postprocedural HE in patients receiving TIPS for refractory ascites.

## 2. Materials and Methods

This single-center retrospective study received institutional review board approval (IRB#10-000464). All study data were stored using REDCap 10.0.1 (Research Electronic Data Capture) within our institution. Demographic information, clinical data, and laboratory data were collected. All adult patients at our institution that received TIPS placement for the primary indication of refractory ascites from 2007 to 2019 were included for analysis. Exclusion criteria included a history of portal vein thrombosis, orthotopic liver transplantation prior to TIPS placement, or being lost to follow-up. CKD stage 3 or higher was defined as having GFR < 60 (unit = mL/min/1.73 m^2^) for at least 3 months prior to TIPS, as is defined by the National Kidney Foundation–Kidney Disease Outcomes Quality Initiative guidelines [13]. All patients were referred for TIPS by hepatologists at our institution, where lactulose is not routinely prescribed for primary prophylaxis of HE. Standardized West Haven Criteria grading was not available for determining the severity of post-TIPS HE. 

### 2.1. Technique 

All TIPS placements were performed using the standard method [14,15]. The Rosch–Uchida transjugular access set (Cook Medical, Bloomington, IN, USA) was used to access the portal vein from the hepatic vein via right internal jugular venous access. Once portal vein access was confirmed, the portosystemic pressure gradient (PSPG) was calculated using portal pressures and right atrial systemic pressures. Evaluation of anatomy and flow dynamics was performed by portovenogram. Following that, the Viatorr endoprosthesis stent–graft (Gore Medical Inc, Flagstaff, AZ, USA) was placed, allowing for a connection between the portal vein and the hepatic vein/inferior vena cava confluent. An appropriately sized balloon (6–10 mm) was used for dilation of the stent–graft, and once the stent–graft was appropriately placed, the post-TIPS PSPG was measured and a final portovenogram was obtained. 

### 2.2. Statistical Methods

Descriptive statistics were used to report baseline characteristics and are represented as percentages as well as mean (SD). Logistic regression models were used to examine GFR at time of TIPS placement as predictors for the development of post-TIPS HE within 60 days for all patients, regardless of renal status. GFR at time of TIPS placement was stratified at cut points every 30 (unit = mL/min/1.73 m^2^), in which each level of stratification was assessed as a potential predictor for post-TIPS HE compared with the GFR ≥ 90 group, with all dialysis patients included in the GFR < 30 group. Logistic regression models were subsequently used to examine GFR at time of TIPS placement, as well as chronic renal status as a potential predictor for the outcome of 60-day post-TIPS HE. Results are presented with odds ratios, *p* values, and 95% confidence intervals. *p* values < 0.05 were regarded as statistically significant. All analyses were performed using Stata/IC 16.1 statistical software.

## 3. Results

Our electronic medical record chart review revealed 218 patients that received TIPSs for refractory ascites at our institution between 2007 and 2019. Among these patients, 17 were excluded based on at least one exclusion criteria, including having portal vein thrombosis (*n* = 8), having received orthotopic liver transplantation prior to TIPS placement (*n* = 3), and being lost to follow up (*n* = 6). A total of 201 adult TIPS recipients (mean age 59.1 (±10.2) years, mean Model for End-Stage Liver Disease (MELD) score 17.3 (±6.9)) were included for analysis (Table 1). In total, 78 (39%) patients met the criteria for CKD based on the 3-month GFR criteria, of which 16 (8%) patients were on dialysis (Figure 1). Among all patients not on dialysis (*n* = 185), mean GFR at time of TIPS placement was 62.7 (±28.2), with a mean GFR at TIPS placement (excluding patients on dialysis) of 40.2 (±12.9) for CKD patients (*n* = 62) and 74.0 (±27.0) for patients without renal insufficiency (*n* = 123). 

Among the entire study population, all 201 patients had a primary indication of refractory ascites for their TIPS placement. A portion of these patients (20%) had a history of other complications of portal hypertension in addition to their refractory ascites, such as esophageal varices, gastric varices, or hepatic hydrothorax. The majority of patients had refractory ascites secondary to cirrhosis (*n* = 195), with a small minority being treated for complications due to noncirrhotic portal hypertension. Etiologies of cirrhosis included hepatitis C (36%), hepatitis B (6%), alcoholic cirrhosis (30%), nonalcoholic steatohepatitis (25%), cryptogenic cirrhosis (9%), and other etiologies (9%), in which 24 patients were noted to have more than one documented cirrhosis etiology.

A total of 101 patients (50%) were found to have post-TIPS HE within 60 days following TIPS placement. Logistic regression analysis was used to examine the role of acute renal function as a potential predictor for post-TIPS HE (Table 2). Compared with patients with GFR ≥ 90, the GFR < 30 group (including all patients on dialysis regardless of GFR) was found to be at significantly increased risk for post-TIPS HE within 60 days (OR, 3.56; 95%CI, 1.19–10.7; *p* = 0.023). GFRs at time of TIPS placements of 30–60 (OR, 1.28; 95%CI, 0.56–2.88; *p* = 0.559) and 60–90 (OR, 0.60; 95%CI, 0.25–1.46; *p* = 0.257) were not found to be significant predictors for post-TIPS HE within 60 days. CKD stage 3 or higher as defined as having a 3-month pre-TIPS GFR < 60 was found to be a significant predictor for our outcome of 60-day post-TIPS HE (OR, 2.52; 95%CI, 1.40–4.53; *p* = 0.002). 

## 4. Discussion

The development of HE is a sequela of both portal hypertension and a complication of TIPS placement, and it has important consequences for the clinical course of patients with decompensated liver disease. The results of this study suggest that underlying renal function is an important risk factor towards the development of HE after TIPS placement for refractory ascites. Among all patients receiving TIPSs for refractory ascites, regardless of chronic renal status, only patients with severely decreased renal function, defined as those with GFR < 30 at time of TIPS placement, or who were noted to be on dialysis at time of TIPS placement, were at increased risk for post-TIPS HE compared with those with normal GFR ≥ 90. Patients with only moderately or mildly decreased GFRs at time of TIPS placement were not found to be at statistically significantly increased risk for the development of post-TIPS HE. Chronic renal dysfunction, or GFR < 60 for at least 3 months prior to TIPS placement, were also found to be strong predictors for post-TIPS HE.

Existing studies examining the role of renal function on the development of post-TIPS outcomes have suggested that TIPSs are safe and effective in patients with renal disease. Haskal and Radhakrishnan examined outcomes among six hemodialysis-dependent TIPS recipients, demonstrating that TIPS could be successfully deployed in patients with end-stage renal disease (ESRD); however, grade 2 or higher HE was noted in all six patients following TIPS placement [11]. A more recent retrospective study by Lakhoo et al. examined clinical outcomes following TIPS placement in 17 patients with advanced CKD [12]. This study supported the safety and efficacy of TIPSs in patients with CKD and suggested moderately increased rates of HE (47%) within their study population. It should be noted that the majority of patients from this study had a primary TIPS indication of variceal hemorrhage, and no patients were hemodialysis-dependent. While these existing studies highlight the potential for increased risk of post-TIPS HE among TIPS recipients with renal dysfunction, they are limited to small-scale retrospective studies with poor heterogeneity, as they neglect to control for TIPS indication. 

TIPS placement can be of significant clinical benefit to patients with complications due to portal hypertension; however, the development of new or worsening HE after TIPS placement is a significant concern. Moreover, managing complications of portal hypertension is important in patients concurrently suffering from renal insufficiency who are found to be at increased risk for the development of HE after TIPS placement. Findings from this study may be incorporated into the evidence-based risk stratification of patients with advanced liver and kidney disease, as well as help with HE screening and overall clinical management of TIPS recipients with complicated clinical histories. 

As the results of this study suggest that renal patients are at increased risk of HE after TIPS placement, particular care should be placed on the postprocedural monitoring and management of such patients. Mainstay pharmacological management for HE includes lactulose and rifaximin, both of which have been documented to potentially infer clinical benefits in the setting of renal disease. In patients with cirrhosis, rifaximin has been demonstrated to increase mean arterial pressure and GFR, while decreasing plasma endotoxin, serum interleukin-6, and serum tumor necrosis factor-α [16]. Moreover, the prolonged use of rifaximin in the setting of cirrhosis has been associated with reduced incidence of acute kidney injury and hepatorenal syndrome [17]. Among patients with CKD, lactulose has been shown to be safe and tolerable for hemodialysis-dependent patients [18], and it has also been shown to decrease circulating levels of urea, creatinine, uric acid, and β2-microglobulin [19,20]. TIPS recipients with underlying renal disease may especially benefit from the use of lactulose and rifaximin, and future studies on the treatment and prevention of HE among these patients should investigate this application of these already widely used pharmacologic therapies, including the role of dosage, as well as their potential utility as primary prophylaxis prior to TIPS placement. 

Further research will be required to thoroughly elucidate the mechanisms by which renal impairment augments the risk of developing post-TIPS HE. Nevertheless, we can hypothesize several potential mechanisms based on our current scientific understanding. The kidneys are known to play a critical role in both ammonia metabolism and excretion [21], and therefore renal impairment could directly lead to hyperammonemia as a result of disturbances in renal ammoniagenesis, as well as impaired ammonia clearance. Additionally, there is growing evidence for the role of systemic inflammation in the pathogenesis of HE [22]. Given that CKD is understood to precipitate a proinflammatory state [23], the relationship between renal dysfunction and HE may derive from a heightened inflammatory response. 

There are several notable limitations to this study. This retrospective study used data from electronic medical records at a single institution, with notable limitations in both generalizability and the potential to ascertain causal associations. This study was also restricted in the control of variables available for analysis; for example, we were unable to examine the severity of post-TIPS HE. Future research would benefit from a prospective cohort study design among patients with concomitant renal and liver disease who may be candidates for TIPS placement for refractory ascites, with more comprehensive assessments of HE severity at predetermined timepoints following TIPS placement. Moreover, our findings must be interpreted in the context of this study, which focused only on the role of renal function on post-TIPS HE. To that end, continued research is necessary to assess the capacity for other known risk factors to modulate post-TIPS HE risk among CKD patients, as well as the potential influence of confounding variables on our findings. Another limitation of this study is the prolonged duration of review, which spanned from 2007 to 2019. During this period of study, there were likely to be some variations in the type of stents used, TIPS placement techniques, and patient selection criteria. Throughout the duration of this study, multiple interventional radiologists would have performed TIPS placement, and the systematic use of controlled-expansion stents was only implemented following 2016. This potential time bias represents a major limitation of our study, which must be considered when interpreting our findings. However, we believe that this potential time bias is at least in part attenuated by our relatively large sample size and that selection criteria for TIPS placement has not changed significantly over the last decade. Nevertheless, these limitations inherent to our dataset must be considered earnestly when interpreting our findings, and broader multicenter studies are recommended to affirm these conclusions. In addition, efforts should be made during future studies to standardize TIPS placement techniques as much as possible.

## 5. Conclusions

Patients with underlying renal impairment that receive TIPSs for managing recurrent ascites are at increased risk for the development of HE. When engaging in the clinical management of patients with renal dysfunction in need of TIPS placement for complications of portal hypertension, particular emphasis should be placed on the potential for decompensation due to encephalopathy during the immediate postprocedural period, in addition to the ensuing weeks to months following TIPS placement. Strategies for monitoring these patients may include increasing the frequency of formal HE evaluation, having a sensitivity towards subtle changes in behavior that may represent early signs of HE, as well as potentially lowering the threshold for implementing HE prophylaxis. However, given the multifactorial nature of HE, all management should be tailored towards each patient’s individual needs and circumstances.

## Figures and Tables

**Figure 1 biomedicines-11-02171-f001:**
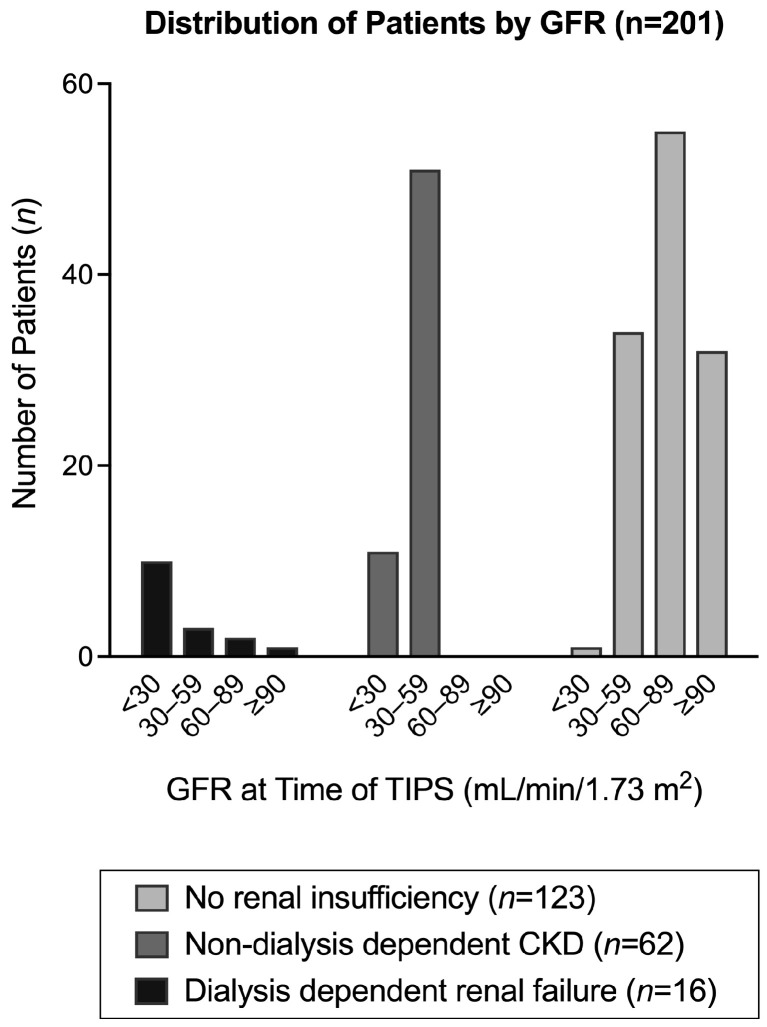
Distribution of patients by glomerular filtration rate (GFR) at time of transjugular intrahepatic portosystemic shunt (TIPS) placement. Abbreviations: GFR, glomerular filtration rate; TIPS, transjugular intrahepatic portosystemic shunt.

**Table 1 biomedicines-11-02171-t001:** Baseline Characteristics for Transjugular Intrahepatic Portosystemic Shunt (TIPS) Placement for Refractory Ascites (*n* = 201).

Characteristic	*N* (%)	Mean (*SD*)
Age, years		59.1 (10.2)
Sex		
Male	123 (61.2)	
Female	78 (38.8)	
Race		
White (non-Hispanic)	107 (53.2)	
Black	2 (1.0)	
Hispanic (any race)	65 (32.3)	
Asian	12 (6.0)	
Other	15 (7.5)	
TIPS Indication		
Ascites only	175 (87.0)	
Ascites and hydrothorax	26 (12.9)	
History of pre-TIPS HE	125 (62.2)	
On dialysis at time of TIPS	16 (8.0)	
GFR (mL/min/1.73 m^2^) at time of TIPS (nondialysis, *n* = 185)		62.7 (28.2)
MELD score at time of TIPS		17.3 (6.9)
MELD-Na score at time of TIPS		20.9 (6.9)
Child–Pugh score at time of TIPS		8.8 (1.3)
Cirrhosis etiology (*n* = 195)		
Hepatitis C	70 (35.9)	
Hepatitis B	12 (6.2)	
Alcoholic cirrhosis	59 (30.3)	
Nonalcoholic steatohepatitis	48 (24.6)	
Cryptogenic	18 (9.2)	
Other	18 (9.2)	
TIPS endoprosthesis type		
Without controlled expansion	149 (74.1)	
Controlled expansion	52 (25.9)	
Pre-TIPS Pressure Measurements		
PV pressure		26.1 (6.1)
RA pressure		9.8 (4.7)
Portosystemic gradient		16.8 (5.3)

Abbreviations: TIPS, transjugular intrahepatic portosystemic shunt; MELD, model for end-stage liver disease; HE, hepatic encephalopathy; PV, portal vein; RA, right atrium.

**Table 2 biomedicines-11-02171-t002:** Logistic Regression Analysis for Post Transjugular Intrahepatic Portosystemic Shunt (TIPS) Hepatic Encephalopathy (HE) by Glomerular Filtration Rate (GFR).

GFR at Time of TIPS (mL/min/1.73 m^2^)	Post-TIPS HE within 60 Days (*n* = 201)			
	*n*	OR	*p* Value	95% CI
<30 or dialysis	29	3.56	0.023	1.19–10.7
30–59	85	1.28	0.559	0.56–2.88
60–89	55	0.60	0.257	0.25–1.46
>90	32	1.00	—	—

Abbreviations: GFR, glomerular filtration rate; TIPS, transjugular intrahepatic portosystemic shunt; HE, hepatic encephalopathy.

## Data Availability

The data presented in this study are available on request from the corresponding author. The data are not publicly available due to privacy restrictions and HIPAA compliance.

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
