# Peer review of "The Impact of Renal Function on Hepatic Encephalopathy Following TIPS Placement for Refractory Ascites"

_biomedicines, 2023, doi:10.3390/biomedicines11082171_

Round 1

Reviewer 1 Report

I have the following commnet

1. Several factors are known to be a risk factors for post-TIPS HE- age, CTP score, MELD, MELD-Na,  change in pressure gradient, prior HE, pleural effusion, high ammonia , stent size etc. These factor have been omitted and presence of CKD is expressed as the sole factor for post-TIPS HE.

2. Try to compare the above mentioned factors between (i) CKD versus no CKD subgroups undegone TIPS (ii) participants with or without post-TIPS HE.

None

Author Response

We appreciate the opportunity to proceed with revising our manuscript in response to feedback from the reviewers and the Clinical and Translational Gastroenterology editorial office. We thank the reviewers for their continued careful review, thoughtful comments, and helpful suggestions, which have enabled us to submit a revised manuscript that has been strengthened considerably. In our response below, we address each of the comments raised by the reviewer in a point-by-point manner. We have incorporated all changes in response to reviewer feedback in the manuscript and we identify where these modifications are located in the manuscript.

Reviewer 1

C1. Several factors are known to be a risk factors for post-TIPS HE- age, CTP score, MELD, MELD-Na,  change in pressure gradient, prior HE, pleural effusion, high ammonia , stent size etc. These factor have been omitted and presence of CKD is expressed as the sole factor for post-TIPS HE. Try to compare the above mentioned factors between (i) CKD versus no CKD subgroups undegone TIPS (ii) participants with or without post-TIPS HE.

R1. We thank the reviewer for their thoughtful review, and agree that the assessment of multiple known risk factors for post-TIPS HE in the setting of CKD and no CKD is an important topic that deserves to be evaluated. The aim of our study was to specifically examine the role of renal function on post-TIPS HE, and further analysis surrounding the aforementioned risk factors would be beyond the scope of this paper. Nevertheless, the reviewer has identified a noteworthy area for continued study, which we have elaborated upon in our discussion section (page 7,  lines 212-215).

Reviewer 2 Report

This paper, entitled "The Impact of Renal Function on Hepatic Encephalopathy Following TIPS Placement for Refractory Ascites", sets out to explore the role of renal function in hepatic encephalopathy (HE) following transjugular intrahepatic portosystemic shunt (TIPS) placement. While the paper highlights some interesting points and provides a relevant study, it does possess numerous critical limitations that must be addressed to fully appreciate its value and impact.

·        One of the primary concerns is the study design. As a retrospective study, it is intrinsically limited in its ability to establish causal relationships. In addition, the use of a single institution's data further limits the generalizability of the study. Furthermore, the authors note that the study did not take into account the severity of post-TIPS HE, which is a crucial factor in evaluating the impact of renal function on HE.

·        Although a sample of 201 TIPS patients for refractory ascites seems substantial, it is critical to observe that this data was collected over 12 years. Also, the study's patient population seems to lack heterogeneity as the data is derived from a single institution. This narrow sample may not capture the wider variability in patient demographic, comorbidities, and other factors that could influence outcomes, thereby hampering its external validity.

·        It appears that the authors primarily relied on logistic regression analyses for their conclusions. However, the study does not mention any control for potential confounders, which could skew the results. It is important to adjust for potential confounders in such an analysis to ensure that the observed associations are not due to some third variable.

·        From a clinical perspective, while the authors' observations could potentially impact the management of patients undergoing TIPS, the paper lacks explicit guidance on how to integrate these findings into clinical practice. For example, the authors recommend additional monitoring of renal patients after TIPS, but do not provide specific strategies or interventions that could be used.

·        The study does not provide a robust pathophysiological explanation for the association between renal function and HE post-TIPS. A deeper exploration into the possible mechanisms would strengthen the study and potentially lead to more targeted interventions.

·        The study spans 12 years, a period during which significant advancements in medical technology and practices could have occurred. Such advancements may influence patient outcomes independently of renal function, and failing to account for this is a major oversight. The authors did acknowledge this limitation but dismissed it based on their relatively large sample size. This dismissal seems inappropriate given the potential impact of this factor on the study's conclusions.

In conclusion, while this study provides a significant step towards understanding the role of renal function in HE following TIPS placement, it is constrained by several weaknesses that limit the impact and application of its findings. Future studies should attempt to address these limitations by adopting a multi-centre, prospective design, involving a larger and more diverse patient population, controlling for potential confounders, and providing more mechanistic insights into the associations observed.

Author Response

We appreciate the opportunity to proceed with revising our manuscript in response to feedback from the reviewers and the Clinical and Translational Gastroenterology editorial office. We thank the reviewers for their continued careful review, thoughtful comments, and helpful suggestions, which have enabled us to submit a revised manuscript that has been strengthened considerably. In our response below, we address each of the comments raised by the reviewer in a point-by-point manner. We have incorporated all changes in response to reviewer feedback in the manuscript and we identify where these modifications are located in the manuscript.

Reviewer 2

C1. One of the primary concerns is the study design. As a retrospective study, it is intrinsically limited in its ability to establish causal relationships. In addition, the use of a single institution's data further limits the generalizability of the study. Furthermore, the authors note that the study did not take into account the severity of post-TIPS HE, which is a crucial factor in evaluating the impact of renal function on HE.

R1. We thank the reviewer for highlighting these inherent limitations to our study. Despite these limitations, we believe this study offers useful insight into the association between impaired renal function and post-TIPS HE, and have emphasized these limitations in our discussion section (page 7, lines 205-206).

C2. Although a sample of 201 TIPS patients for refractory ascites seems substantial, it is critical to observe that this data was collected over 12 years. Also, the study's patient population seems to lack heterogeneity as the data is derived from a single institution. This narrow sample may not capture the wider variability in patient demographic, comorbidities, and other factors that could influence outcomes, thereby hampering its external validity.

R2. We appreciate your attention to the potential limitations of our study and recognize the concern about our sample size, data collection period, and population heterogeneity. We agree that a larger sample size over a shorter collection period would strengthen the study, but also recognize that this single-center data is derived from a large volume academic center, and the sample size remains reasonable given the specified criteria (TIPS placement for the primary indication of refractory ascites). Nevertheless, we do agree that the external validity of our findings is constrained due to these limitations, and in the revised manuscript have emphasized that these findings should therefore be interpreted cautiously, and affirm the need for broader multi-center studies in order to provide additional validation of these insights (page 7, lines 212-215).

C3. It appears that the authors primarily relied on logistic regression analyses for their conclusions. However, the study does not mention any control for potential confounders, which could skew the results. It is important to adjust for potential confounders in such an analysis to ensure that the observed associations are not due to some third variable.

R3. We thank the reviewer for noting this important point regarding our methodology. In the design of this study, we focused on presenting unadjusted associations derived from the logistic regression models, and acknowledge that these findings could be subject to confounding. The intention of this study was to highlight these unadjusted relationships, and fully recognize the need for future studies to adjust for confounders in order to better understand causal relationships. As above, we have amended our discussion to more clearly communicate our approach and its implications (page 7, lines 212-215).

C4. From a clinical perspective, while the authors' observations could potentially impact the management of patients undergoing TIPS, the paper lacks explicit guidance on how to integrate these findings into clinical practice. For example, the authors recommend additional monitoring of renal patients after TIPS, but do not provide specific strategies or interventions that could be used.

R4. We acknowledge the need for more explicit guidance on recommendations for integrating these findings into clinical practice. Our revised manuscript now presents more specific guidance on strategies such as careful monitoring for post-TIPS HE during the immediate post-procedural period, in addition to the ensuing weeks to months following TIPS placement. Such strategies may include include increasing the frequency of formal HE evaluation, having a sensitivity towards subtle changes in behavior that may represent early signs of HE, as well as potentially lowering the threshold for implementing HE prophylaxis. However, given the multifactorial nature of HE, all management should be tailored towards each patient’s individual needs and circumstances. These points have been amended to our revised discussion (page 7, lines 233-239).

C5. The study does not provide a robust pathophysiological explanation for the association between renal function and HE post-TIPS. A deeper exploration into the possible mechanisms would strengthen the study and potentially lead to more targeted interventions.

R5. Thank you for bringing up this important point about presenting a pathophysiological explanation. A discussion of proposed mechanisms for the association between renal function and post-TIPS HE has been added to our discussion section, and include disturbances to renal ammoniagenesis, impaired renal ammonia clearance, as well as systemic inflammation as a result of CKD (page 6-7, lines 194-203).

C6. The study spans 12 years, a period during which significant advancements in medical technology and practices could have occurred. Such advancements may influence patient outcomes independently of renal function, and failing to account for this is a major oversight. The authors did acknowledge this limitation but dismissed it based on their relatively large sample size. This dismissal seems inappropriate given the potential impact of this factor on the study's conclusions.

R6. Thank you for your thoughtful review, and noting the important potential impact of technological and procedural evolution over the course of our study period. Although we did acknowledge this limitation, we have not sufficiently emphasized this potential impact. Moreover, we agree that the relatively large sample size for this study does not negate this genuine concern, and apologize if the previous version of our manuscript suggested otherwise. We have amended our discussion to better highlight this limitation (page 7, lines 220-223, 224-228).

Round 2

Reviewer 1 Report

Thanks

Reviewer 2 Report

The authors have revised well. The manuscript can be accepted for publication.